# Potential for online crowdsourced biological recording data to complement surveillance for arthropod vectors

Benjamin Cull*

Department of Entomology, University of Minnesota, St. Paul, Minnesota, United States of America

* cull0122@umn.edu

## Abstract

Voluntary contributions by citizen scientists can gather large datasets covering wide geographical areas, and are increasingly utilized by researchers for multiple applications, including arthropod vector surveillance. Online platforms such as iNaturalist accumulate crowdsourced biological observations from around the world and these data could also be useful for monitoring vectors. The aim of this study was to explore the availability of observations of important vector taxa on the iNaturalist platform and examine the utility of these data to complement existing vector surveillance activities. Of ten vector taxa investigated, records were most numerous for mosquitoes (Culicidae; 23,018 records, 222 species) and ticks (Ixodida; 16,214 records, 87 species), with most data from 2019–2020. Case studies were performed to assess whether images associated with records were of sufficient quality to identify species and compare iNaturalist observations of vector species to the known situation at the state, national and regional level based on existing published data. Firstly, tick data collected at the national (United Kingdom) or state (Minnesota, USA) level were sufficient to determine seasonal occurrence and distribution patterns of important tick species, and were able to corroborate and complement known trends in tick distribution. Importantly, tick species with expanding distributions (*Haemaphysalis punctata* in the UK, and *Amblyomma americanum* in Minnesota) were also detected. Secondly, using iNaturalist data to monitor expanding tick species in Europe (*Hyalomma* spp.) and the USA (*Haemaphysalis longicornis*), and invasive *Aedes* mosquitoes in Europe, showed potential for tracking these species within their known range as well as identifying possible areas of expansion. Despite known limitations associated with crowdsourced data, this study shows that iNaturalist can be a valuable source of information on vector distribution and seasonality that could be used to supplement existing vector surveillance data, especially at a time when many surveillance programs may have been interrupted by COVID-19 restrictions.

## Introduction

Researchers are increasingly making use of data collected by citizen scientists, either through participatory projects or using smartphone apps. Enlisting the help of the public to gather

**Funding:** Supported in part by grants from the National Institutes of Health (nih.gov), R01AI042792 and R01AI049424 to UG Munderloh. The funders had no role in study design, data collection and analysis, decision to publish, or preparation of the manuscript.

**Competing interests:** The authors have declared that no competing interests exist.

scientific information can result in large datasets covering large geographical areas, as well as educating participants about the scientific process and the topic addressed by the project [1]. Vector surveillance has successfully used citizen science to collect data on arthropod species' distribution, seasonality and human/animal exposure, with a number of passive surveillance projects implemented to improve knowledge on regional vector occurrence. For example, these include projects to monitor ticks [2–9], mosquitoes [10, 11], and triatomines [12], and typically involve the submission of vector specimens or photos and associated information (such as location, host, date) to researchers for identification and documentation. A major benefit of the use of citizen science in vector surveillance is the ability to improve the participants' knowledge of vector avoidance behaviors, potentially reducing the incidence of vector-borne disease. For example, mosquito projects may make participants aware of mosquito breeding sites around the home and prompt their removal, which benefits the entire community by reducing biting nuisance and risk of mosquito borne disease. Once set up, citizen science projects can be continuously run at relatively low cost, making them a resource-efficient method of collecting large-scale vector data when compared to traditional active surveillance methods that require more time and resources to perform, particularly when attempting to cover large geographical areas.

The increasing use of smartphones has led to the development of apps that allow members of the public to report sightings of, and/or exposure to, potential vector species directly to public health professionals and researchers. Smartphone apps such as Tekenbeet in the Netherlands [13], TekenNet/TiquesNet in Belgium (https://epistat.wiv-isp.be/ticks/), Switzerland's Tick Prevention (https://zecke-tique-tick.ch/en/app-tick/), France's Signalement Tique (https://www.citique.fr/), and The Tick App [14] and TickTracker (https://ticktracker.com/) in the United States enable citizens to report tick exposure, view current tick activity in their region, and access public health information on correct tick removal and tick-borne disease symptom recognition. The MosquitoAlert app, developed in Spain, has been highly effective in aiding public health officials to track the spread and establishment of the invasive mosquito *Aedes albopictus*, and also helped to detect *Aedes japonicus*, another invasive mosquito species of public health concern [15, 16].

iNaturalist (www.inaturalist.org) is a joint initiative of the California Academy of Sciences and the National Geographic Society supporting a growing online community of naturalists who upload georeferenced photos of wildlife sightings via a smartphone app. However, not all photos are taken with smartphones and there are some very high quality images captured with digital and microscope cameras. Upon submission of an observation, image-based recognition suggests an identification based on similar images that have been verified, and then other users can suggest or confirm identifications. Each observation has its own page where users can discuss the record and add annotations such as sex and life stage. Once multiple agreeing identifications have been made, the record becomes "research grade" and is uploaded to scientific data repositories such as the Global Biodiversity Information Facility (gbif.org), making the data more widely available. In mid-July 2020, the iNaturalist community consisted of over 2.86 million users with contributions totaling over 43.7 million observations of 286,260 species around the world. By early August 2020 there were over 2.96 million users and 45.8 million observations of 289,800 species, and by mid-September the numbers were 3.14 million users with 49.4 million observations of 295,517 species. These numbers indicate a rapidly growing repository of biological records from across the globe.

Data from the iNaturalist platform are starting to be used by researchers for a variety of applications, and have been utilized to study urban biodiversity [17], confirm new country reports of the invasive ladybird *Harmonia axyridis* in Latin America [18], monitor urban red fox and coyote populations [19], aid mapping of global termite distribution and diversity [20], record the phenology of flowering in plants [21], monitor fish for signs of black spot disease

[22], and study the distribution, phenology, and host plant associations of the queen butterfly *Danaus gilippus thersippus* [23]. Observations from iNaturalist could also be useful for the study of vector species of importance to public and veterinary health, but this potential has yet to be explored. A major advantage of this platform over existing citizen science surveillance projects is that the recording platform already exists, with data readily available and continually contributed, and therefore using it requires no resources except the time needed to search, identify records and analyze data. Here the iNaturalist data available for important vector taxa were assessed, and case studies were performed to examine how these data could be used to obtain additional data on vector populations as a means to complement existing vector surveillance approaches. This study also aims to raise awareness of iNaturalist as a source of additional vector records for those involved in vector surveillance.

## Methods

### Records of taxa containing important vector species

Selected taxa containing major vector species of medical and/or veterinary importance [24] were chosen for inclusion in the analysis. These included: mosquitoes (Family Culicidae), containing vectors of pathogens causing malaria, filariasis, and viral diseases such as chikungunya, dengue, Rift valley fever, West Nile, yellow fever, and Zika; ticks (Order Ixodida), containing vectors of *Anaplasma* spp., *Babesia* spp., *Borrelia* spp., *Ehrlichia* spp., *Rickettsia* spp., *Theileria* spp., and tick-borne viruses including those causing Crimean Congo hemorrhagic fever, tick-borne encephalitis, Powassan encephalitis and African swine fever; black flies (Family Simuliidae) which transmit filarial nematodes that cause mansonellosis and human and bovine onchocerciasis; kissing bugs (Subfamily Triatominae), vectors of the causative agent of Chagas disease *Trypanosoma cruzi*; fleas (Order Siphonaptera), vectors of *Rickettsia typhi* (endemic typhus) and *Yersinia pestis* (plague); sucking lice (Superfamily Anoplura), containing the human body louse which is a vector of *Rickettsia prowazekii* (epidemic typhus), *Borrelia recurrentis* (louse-borne relapsing fever) and *Bartonella quintana* (trench fever); biting midges (Genus Culicoides) which act as vectors of viruses causing Oropouche fever, Bluetongue disease, Epizootic Hemorrhagic disease, African horsesickness and Schmallenberg, and filarial nematodes causing mansonellosis and equine onchocerciasis; tsetse flies (Family Glossinidae), vectors of protozoan parasites of the genus *Trypanosoma* that cause African sleeping sickness in humans and Nagana in livestock; trombiculid mites (Family Trombiculidae) which contain vectors of *Orientia tsutsugamushi* (scrub typhus); and sand flies (Subfamily Phlebotominae), vectors of *Leishmania* spp. (leishmaniasis), *Bartonella bacilliformis* (Carrion's disease) and Phleboviruses (sand fly fever). Searches were performed on the 'Explore' page of the iNaturalist website for each of these vector groups, and the overall number of observations, number of species, and annual number of observations were recorded. Whilst many species encompassed by some of the taxa above are not vectors, it was assumed that many records in the database were yet to be identified and so these general searches were considered appropriate to capture vector species of interest. Due to the volume of records, they were not individually verified for correct identification or duplication; the analysis served merely as an approximation of the number of records of taxa containing important arthropod vectors available in iNaturalist.

### Case studies to assess the ability of iNaturalist data to complement vector surveillance

iNaturalist records are typically opportunistic observations of wildlife and include an image of the observation, the date and time of the observation, and geolocation data. Records were

obtained by searching the 'Explore' page of the iNaturalist website for the taxon and geographical region of interest, as described in the Results for each case study. Data were downloaded directly from www.inaturalist.org and images associated with each record were identified by an entomologist trained in the morphological identification of ticks and mosquitoes, with reference to the following sources where necessary: European tick species [25]; ticks in North America [26, 27]; invasive mosquitoes [28, 29]. Example images from each case study are shown in S1 Fig. Any records that were unidentifiable, duplicated or misidentified as the taxon of interest were removed before further analysis. Records were mapped using ArcGIS online (ESRI, California) using open-source layers from Natural Earth (naturalearthdata.com).

The potential of iNaturalist data to be able to complement existing vector surveillance was assessed in case studies of a national or state surveillance program, as well as more extensive regional surveillance programs targeting key invasive arthropod vector species. Case studies focused on ticks and mosquitoes both because of the availability of iNaturalist records for these groups and their high medical and veterinary importance. Each of the five case studies was assessed against the following criteria:

1. Are images associated with iNaturalist observations of sufficient quality to be able to identify to genus or species, and therefore identify arthropod vectors of medical/veterinary concern?

2. Are the number of observations sufficient to be able to describe the distribution and seasonality of a species, and compare with the known situation in the region?

3. Is iNaturalist sensitive enough to be able to detect unusual or expanding species, and add new records of an invasive/expanding species outside its known range?

Data on the geographical distribution and seasonality of vector species obtained from iNaturalist were compared to existing data from scientific publications and publicly accessible websites of government and vector control agencies responsible for vector surveillance. Good agreement between these sources, and an ability of iNaturalist to add records of unusual or invasive/expanding species, was taken as evidence supporting the utility of iNaturalist as a method of obtaining additional data for vector surveillance.

## Results

### A. Evaluation of records of taxa containing important vector species

The number of records of vectors of importance for human and animal health in the iNaturalist database were examined. Searches were performed in iNaturalist for each of the vector groups, but records were not verified for correct identification. Mosquitoes (Family Culicidae) had the greatest number of observations, followed by ticks (Order Ixodida), with both groups having thousands of records (Fig 1A) from across six continents. Taxa of other vector groups were less well represented (Fig 1A), and this may be due to a variety of factors including limited geographical range, occurrence in less-developed areas of the world where smartphone use is less common or where iNaturalist is less well known, and/or difficulty in capturing photographs due to small size.

Analysis of annual observations show that their number has been increasing over time, with records substantially higher in 2018, 2019 and 2020 than in previous years (Fig 1B), likely as a result of increasing awareness and use of the iNaturalist platform. By July 2020, the number of observations of mosquitoes, ticks and black flies had already exceeded the total number of observations for each taxon made in the previous year (Fig 1B).

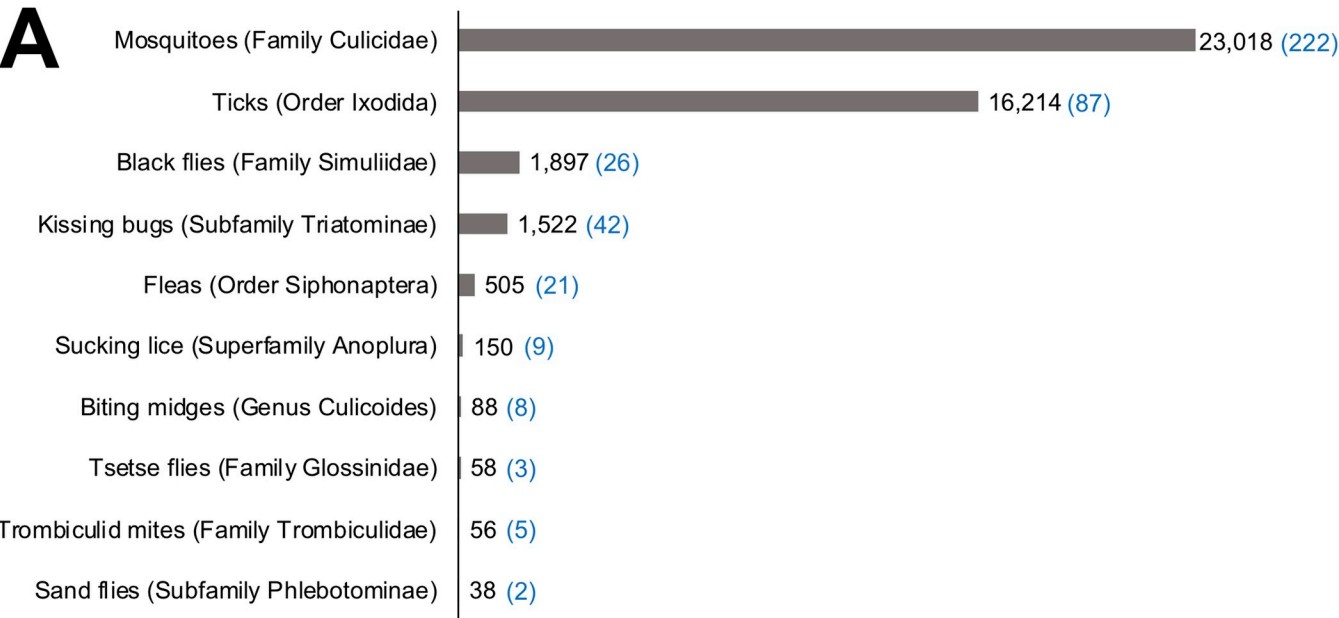

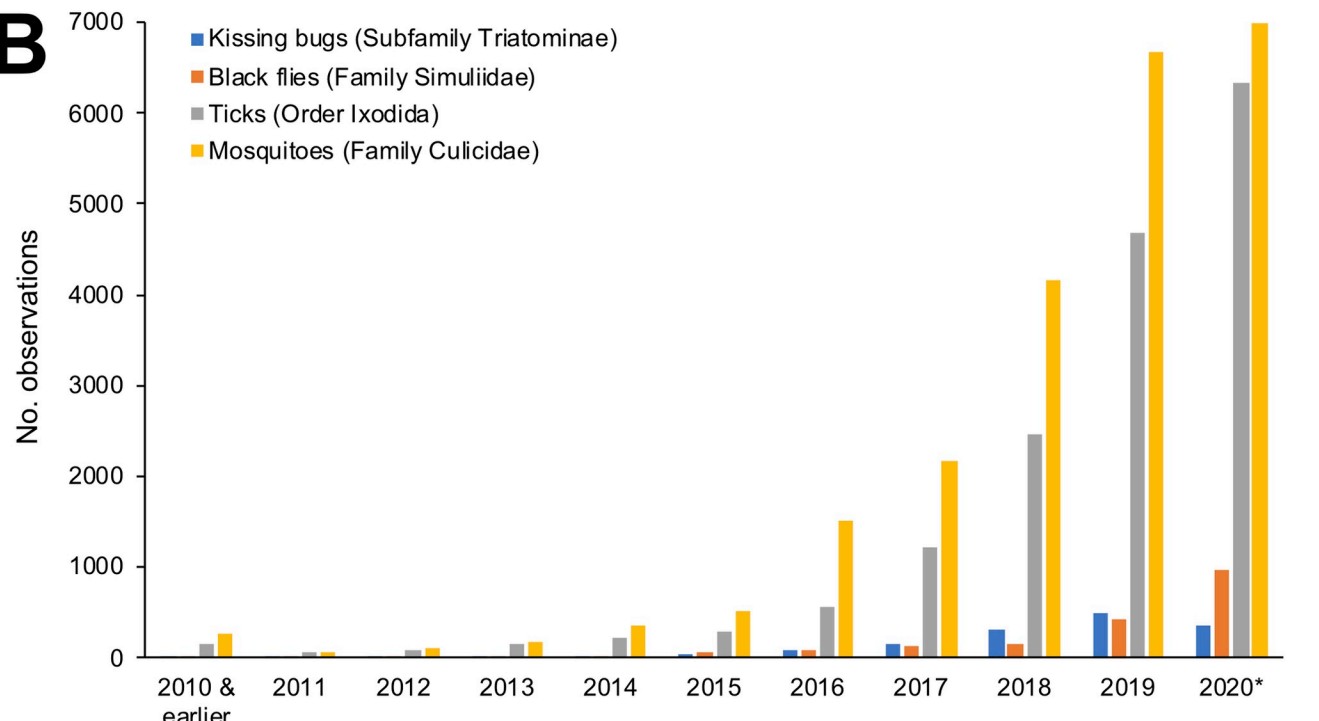

**Fig 1. iNaturalist observations of vector taxa important for human and/or veterinary health.** (A) Number of records of each vector taxon. Figures show the no. observations (black) and the number of species identified (blue in brackets) as of 15 July 2020. (B) Number of observations of the most numerous vector taxa by year. *Data complete up to 15 July 2020.

## B. Vector surveillance at a national or regional level

The potential for data from iNaturalist to be used to supplement vector surveillance at a country or state level was assessed. Local and regional vector surveillance is vital to vector-borne

disease risk assessment, as knowledge of the presence, distribution and seasonal activity of vectors is key for local veterinary and public health organizations to determine whether there is a risk of vector-borne disease transmission, and when and where the risks may be greatest. This information can be used to target surveillance resources and to inform health messaging on vector avoidance and disease recognition. The following two case studies compare data from iNaturalist with the current known situation on ticks in two different regions to examine whether these data are able to complement existing surveillance and provide additional evidence of expanding tick species.

**Case study 1: Tick surveillance in the United Kingdom.**   The United Kingdom (UK) is home to 20 endemic tick species [30, 31], but the species of primary importance for human and animal health is *Ixodes ricinus* (castor bean tick). This species is found throughout the UK, with highest prevalence across southern England and Scotland [3, 32], and is the vector of *Borrelia burgdorferi* s.l. (Lyme borreliosis), the most prevalent arthropod-borne disease in Europe. Whilst human Lyme borreliosis incidence in the UK is low compared to other European countries [33], reported cases increase each year [34, 35]. *Ixodes ricinus* also transmits *Anaplasma phagocytophilum*, *Babesia divergens* and Louping ill virus, which primarily affect livestock in the UK [36]. Recently, tick-borne encephalitis (TBE) virus has been detected in UK populations of *I. ricinus* [37, 38] and the first UK-acquired human TBE case was reported shortly afterwards [39]. A second human TBE case and the second ever UK case of human babesiosis were reported in July 2020 [40]. Other tick species of concern are *Dermacentor reticulatus* and *Haemaphysalis punctata*, both of which are localized to certain areas of the UK and primarily associated with livestock [41, 42]. However, *D. reticulatus* has been linked to cases of canine babesiosis [43] in Essex (east England), and *H. punctata* has recently expanded its range in southeast England with increased reports of human biting and the occurrence of disease and deaths in sheep flocks due to a combination of tick pyemia, babesiosis and theileriosis [42, 44]. Most other tick species present in the UK are specialist ticks of wildlife and therefore rarely encountered by humans and their companion animals. Those most often seen include: *I. hexagonus*, a common parasite on hedgehogs as well as cats and dogs; *I. canisuga*, which feeds mainly on mustelids but also dogs; and *I. frontalis*, a tick primarily associated with birds that occasionally bites companion animals and humans [3, 45].

Records of ticks from the UK were downloaded directly from iNaturalist on 20 July 2020 by searching for all observations of ticks (Order Ixodida) in the UK. Each record was checked to verify identification. In total there were 128 records, and of these 10 were excluded because the specimens were not ticks (four records), the images were duplicates (three records) or the image was not clear enough to identify whether the animal was a tick or not (three records). Eighty six percent of tick records (102/118) could be confidently identified as *Ixodes ricinus*, and in most cases the life stage could also be determined (18 males, 62 females, 26 nymphs, 1 larva; some records included multiple individuals of the same species). Other species identified included *I. hexagonus* (n = 2, 1.7%), *I. frontalis* (n = 2, 1.7%), and *Haemaphysalis punctata* (n = 5, 4.2%). Five records (4.2%) could be identified to genus *Ixodes*, one (0.8%) to genus *Dermacentor*, and one could not be identified to genus (the picture was blurry, but appeared to be either *Haemaphysalis* or a male *Ixodes*).

Observations of *I. ricinus* were distributed across the UK, but with higher numbers from southern England and Scotland (Fig 2A), consistent with the tick's known distribution. All records of *H. punctata* (including the possible record that could not be confirmed due to image quality) were reported within this species' known range between Shoreham-by-Sea and Eastbourne in southeast England (Fig 2A). Interestingly, the record of *Dermacentor* was located outside of the known distribution range of *D. reticulatus* [41], and a closer look at the geography determined that the habitat where the tick was observed is similar to the grazing marshes

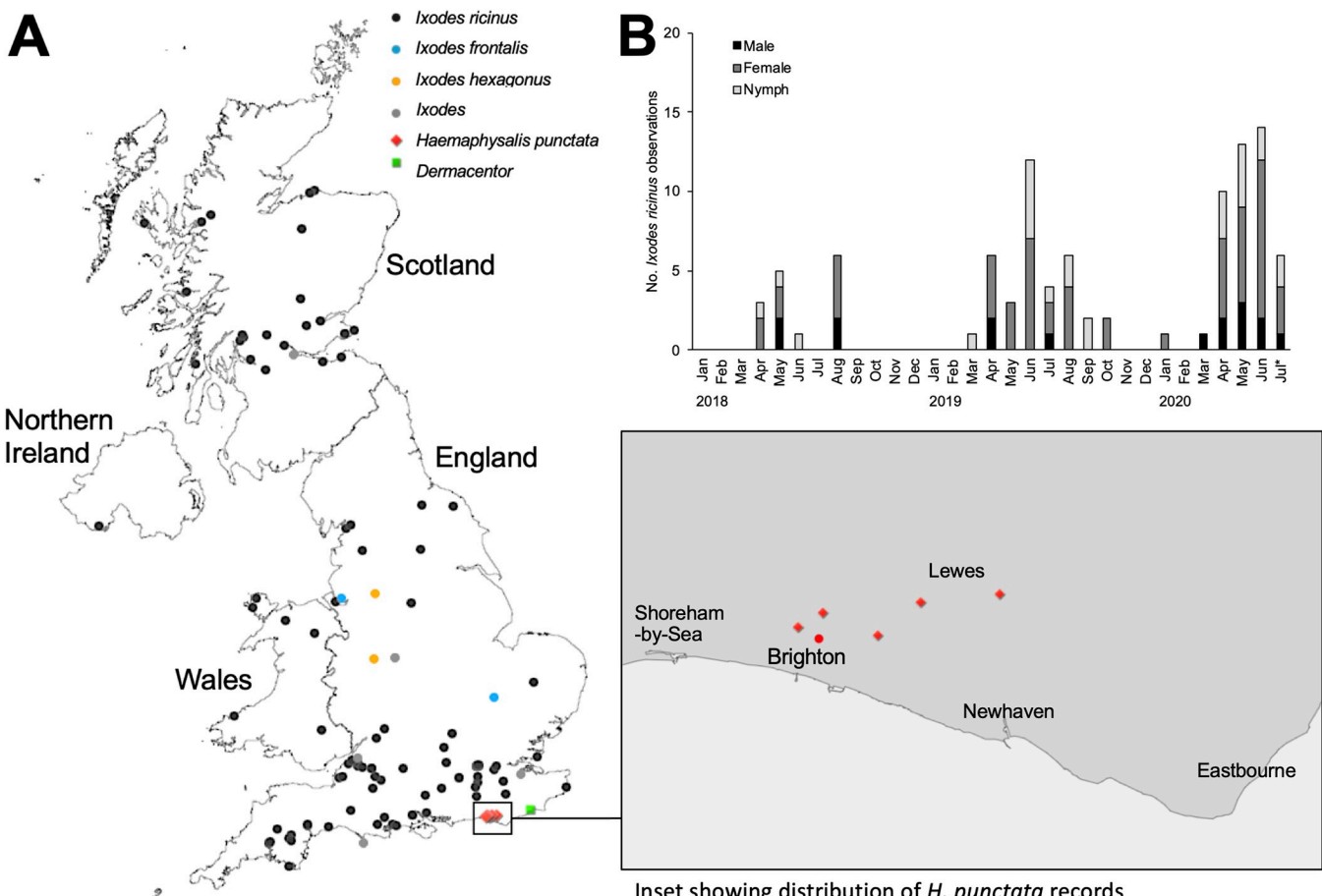

**Fig 2. Summary of iNaturalist records of ticks in the United Kingdom.** (A) Distribution of tick records by genus/species. Inset shows detection of a known localized population of *Haemaphysalis punctata* in southeast England (see Medlock et al., 2018); red circle shows potential *H. punctata* record. (B) Seasonality of *Ixodes ricinus* (separated by life stage) recorded in the UK by iNaturalist users 2018–2020 (*complete to 20 July 2020).

where it is established in Essex; therefore it is possible that the species has been introduced to this area by the movement of infested livestock between sites.

The seasonality of *I. ricinus* records collected by iNaturalist users was also examined, using only those years where greater than 10 observations were recorded (Fig 2B). The highest numbers of *I. ricinus* observations were reported April to August, which reflects the known seasonality of questing *I. ricinus* and human/companion animal tick exposure in the UK [3, 45, 46].

**Case study 2: Tick surveillance in Minnesota, USA.** In Minnesota, the tick of highest public health importance is *Ixodes scapularis* (black-legged tick). This tick has expanded considerably in Minnesota in recent years with the number of counties with established populations of *I. scapularis* increasing from nine to forty-five between 1996 and 2015 [47], accompanied by associated increases in *I. scapularis*-borne disease incidence [48]. All seven human pathogens known to be transmitted by *I. scapularis* in the United States (*Anaplasma phagocytophilum*, *Babesia microti*, *Borrelia burgdorferi*, *Borrelia mayonii*, *Borrelia miyamotoi*, *Ehrlichia muris eauclairensis*, and Powassan virus) have been detected in host-seeking ticks collected in Minnesota [49] and human cases of the diseases caused by these pathogens are reported in Minnesota residents each year [50]. Minnesota has among the highest incidence rates for Lyme borreliosis, anaplasmosis, babesiosis and Powassan virus disease in the US, whilst both *B. mayonii* and *E. muris eauclairensis* appear to be restricted to Minnesota and

Wisconsin. The density of *I. scapularis* is greatest in sites near the Minneapolis-St. Paul metropolitan area, and other high-density sites trend in a northwesterly direction from this location [51], consistent with the distribution of deciduous forest cover in the state, which provides a suitable habitat for the tick and its natural hosts [48, 51].

*Dermacentor variabilis* (American dog tick) is the most commonly encountered tick in the state and is found throughout Minnesota in grassy and wooded areas. This species is rarely involved in pathogen transmission but is occasionally responsible for cases of Rocky Mountain spotted fever (caused by *Rickettsia rickettsii*) and tularemia (*Francisella tularensis*) in Minnesota. Another tick species of concern in Minnesota is *Amblyomma americanum* (Lone star tick), which is currently undergoing an expansion in North America [7, 52–54] and is a vector of several pathogens of public health and veterinary concern, including *Ehrlichia chaffeensis* (human monocytic ehrlichiosis), *E. ewingii* (granulocytic ehrlichiosis), and Heartland virus, as well as a potential vector of Bourbon virus. This tick species is also linked to alpha-gal syndrome and Southern tick-associated rash illness (STARI). Although *A. americanum* is not known to be established in Minnesota, there are sporadic reports and 40 lone star ticks have been documented across the state between 1998 and 2019 by state surveillance, with multiple reports per year during the last decade [55, 56].

A search of the iNaturalist database for ticks (Order Ixodida) in Minnesota on 27 July 2020 came up with a total of 236 observations, with the earliest record from 2008. All records were checked to verify identification. Three observations were excluded because one was a beetle, one was a spider, and the third could not be identified due to poor image quality. Two tick records (0.9%) could not be identified beyond family Ixodidae due to low clarity of the associated image. The majority of the remaining ticks were *D. variabilis* (73.0%;170/233), whilst *I. scapularis* accounted for 58 records (24.9%). Notably, there were two observations of *Amblyomma americanum* (0.9%) recorded in June and July of 2020. There was also one record (0.4%) of *Carios kelleyi*, found in a house; this argasid tick is a common parasite of bats and is found throughout North America, but may infest human dwellings and bite residents resulting in skin lesions [57]. Novel spotted fever group *Rickettsia* species and a relapsing fever group *Borrelia* have been detected in *C. kelleyi*, suggesting some potential for zoonotic transmission [58].

Tick records were distributed throughout much of the state but are clearly aggregated around the Minneapolis-St. Paul metropolitan area where the majority of the state's human population reside (Fig 3A). Fewer observations appear to come from the southern and southwestern parts of the state, which is primarily farmland and prairie habitat and therefore of low suitability for ticks [59]. Observations of *D. variabilis* were located in 42 (48.3%) of the state's 87 counties. Observations of *I. scapularis* from iNaturalist were located in 22/87 (25.3%) counties, and 21/61 (34.4%) counties where this tick species has previously been reported (Fig 3B); however, one observation was from Grant county, where this species is so far not recorded (Fig 3A and 3B), although it is known to be established in the neighboring counties of Douglas (to the east) and Otter Tail (to the north).

The seasonality of tick observations collected by iNaturalist users in years with greater than 10 records show peak activity of *D. variabilis* during April to July and that of *I. scapularis* April to June with a second period of activity observed in October of some years (Fig 3C). These patterns of seasonal activity are consistent with those observed for these tick species in other northern regions of North America [6, 61–67].

## C. Monitoring the invasion or expansion of important vector species

Another potential use of the georeferenced records in the iNaturalist database is aiding the monitoring for invasive or expanding vector species outside their established range. Obtaining

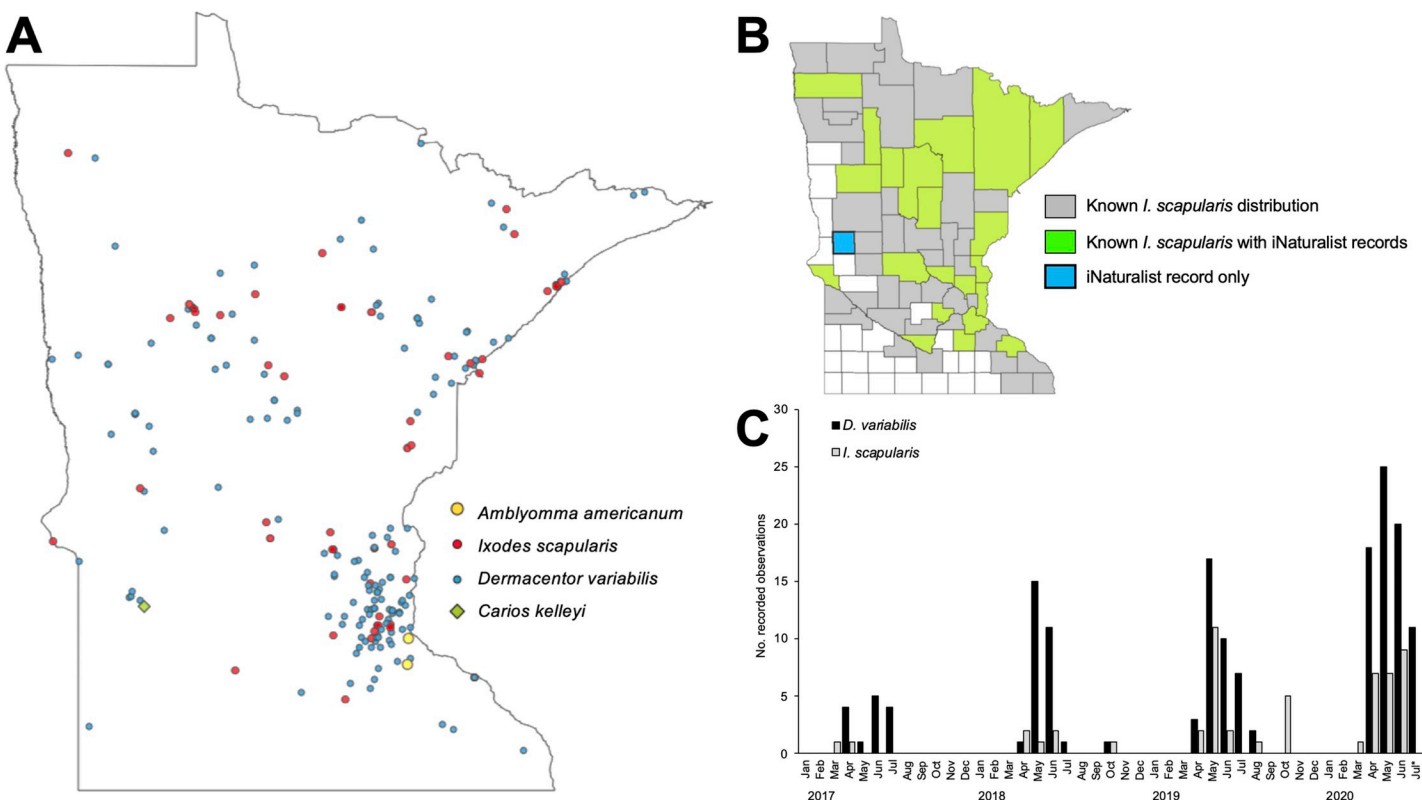

**Fig 3. Summary of iNaturalist records of ticks in Minnesota, USA.** (A) Distribution of tick records by species. (B) Comparison of *Ixodes scapularis* distribution data from iNaturalist with the current known county-scale distribution in Minnesota (as of July 2019, map created based on data from Minnesota Department of Health Vectorborne Diseases Unit [60]). (C) Seasonality of *Dermacentor variabilis* and *Ixodes scapularis* recorded in Minnesota by iNaturalist users 2017–2020 (*data complete to 27 July 2020).

early reports of invasive species before they can establish may enable vector control units to eliminate vectors before their populations become large enough to cause a risk to human and animal health through nuisance biting and/or pathogen transmission. Citizen science has already contributed to the detection of invasive species (e.g. [15]), and can complement existing surveillance aiming to intercept species of concern or monitor their spread once established.

**Case study 3: Monitoring the distribution of *Hyalomma* spp. ticks in Europe.** Ticks of the genus *Hyalomma* are vectors of several human and animal pathogens, including *Theileria* spp., *Babesia* spp., *Anaplasma* spp., *Rickettsia* spp., and *Coxiella burnetti*. Most importantly from a public health perspective, *Hyalomma* spp. are also the vectors of Crimean Congo haemorrhagic fever (CCHF) virus [68], which can cause severe disease in humans with a case fatality rate of up to 40%. In Europe the most widespread *Hyalomma* species, and the primary CCHF virus vector, is *Hy. marginatum* (Fig 4A). Outbreaks of CCHF are mainly reported in southeastern Europe, with the majority of cases in Bulgaria, Russia, and Turkey. The first autochthonous CCHF case was recently reported in Spain [69]. Immature stages of *Hyalomma* are frequently detected on migratory birds entering Europe from Africa [70–75], and it is thought that these could lead to established populations of *Hyalomma* in new areas if nymphs were to drop off at sites with suitable climate and host availability. Indeed, this is thought to be the most likely explanation for the recent establishment of *Hy. marginatum* in southern France [76], an important overwintering stop for migrating birds. There have been a number of adult

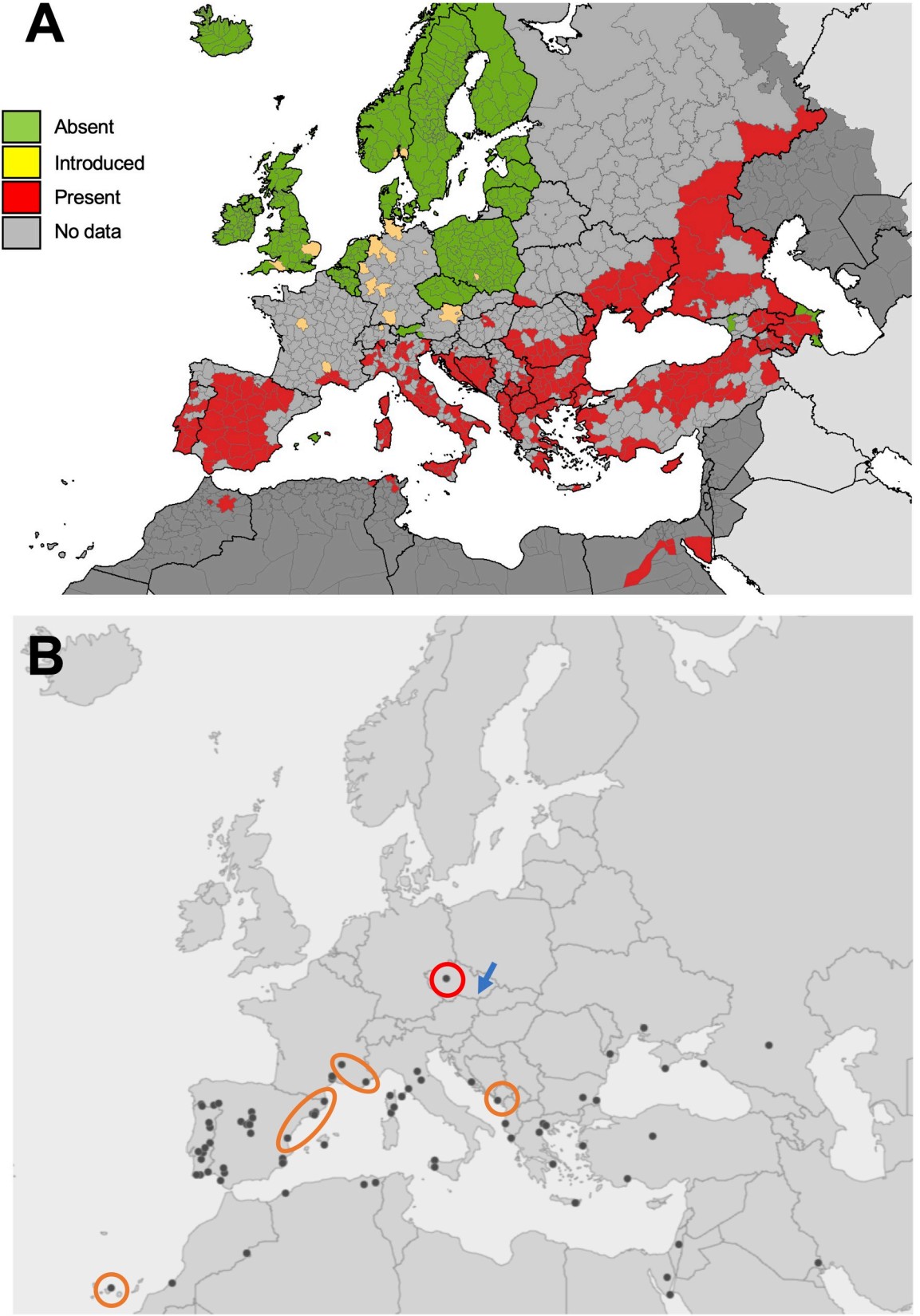

**Fig 4. Distribution of *Hyalomma* records in Europe and neighboring territories.** (A) Known distribution of *Hyalomma marginatum* in Europe as of May 2020. Reprinted from European Centre for Disease Control and Prevention VectorNet project www. ecdc.europa.eu/en/disease-vectors/surveillance-and-disease-data/tick-maps, original copyright ECDC [2020]. (B) Distribution of records of genus *Hyalomma* in Europe from iNaturalist. Red circle highlights an observation of *Hyalomma* in Czechia recorded in 2018; blue arrow shows location of an adult *Hyalomma rufipes* detected in Czechia in 2019 (Hubálek et al., 2020). Records highlighted by orange rings represent potential areas of expansion of *Hyalomma marginatum* in France and Spain, and *Hyalomma* records in the Canary Islands and Montenegro.

*Hy*. *marginatum* or *Hy*. *rufipes* detected in temperate areas of northern and central Europe in recent years, in Austria [77], Czechia [78], Germany [79–81], Hungary [82], the Netherlands [83, 84], and the United Kingdom [85, 86]. These findings of adult *Hyalomma*, which have primarily been reported on large animals and humans, are supportive of the introduction of nymphs that successfully molted under favorable climatic conditions; the detection of so many adults during 2018 is associated with particularly warm temperatures that year [81, 85, 86]. Whilst no CCHF virus was found in any of these ticks, many of the *Hyalomma* adults tested positive for the human pathogen *Rickettsia aeschlimannii* [77, 81, 85, 86], which has important implications for public health.

Observations of *Hyalomma* spp. in Europe were downloaded directly from iNaturalist using the search terms '*Hyalomma*' in 'Europe' on 23 July 2020. All observations (n = 101) were verified to genus level, but due to the difficulty of differentiating *Hyalomma* species from photos without a close look at morphological features, attempts to identify ticks to species level were not made. Consistent with the genus' known distribution in Europe and neighboring regions, records ranged across southern Europe, North Africa and the Middle East (Fig 4). One notable record is that of a *Hyalomma* adult male recorded in Czechia, Central Europe, in October 2018 to the southwest of Prague (Fig 4B, red circle). Although no investigations were made into the record, assuming this observation is accurate it would precede the first published detection of adult *Hyalomma* in the country by one year, when an adult *Hy. rufipes* was found on a horse in the South Moravia region of Czechia (Fig 4B, blue arrow) in October 2019 after presumably being imported on a migratory bird as a nymph and subsequently molting to an adult [78]. Additionally, potential areas of *Hy. marginatum* expansion were identified in the Mediterranean regions of France and Spain (Fig 4B), and whilst the observations were not confirmed as *Hy. marginatum*, it is plausible because this is the most widely distributed *Hyalomma* species in this area and confirmed populations are known to exist in neighboring provinces (Fig 4A); however, *Hy. lusitanicum* also exist in these areas [25]. The observation of *Hyalomma* in the Canary Islands (Fig 4B) is likely to be *Hy. lusitanicum*, which is an abundant species on the archipelago [87]. Another observation that differs from the known *Hy. marginatum* distribution in Fig 4A is from Montenegro (Fig 4B); however, both *Hy. marginatum* and *Hy. scupense* have previously been recorded in this country [25].

**Case study 4: Monitoring populations of the invasive tick *Haemaphysalis longicornis* in the US.** *Haemaphysalis longicornis* is a tick species native to eastern Asia, that has been introduced and established in Australia, New Zealand, and several Pacific Islands. It is a vector of several human and animal diseases, and has the ability to reproduce parthenogenetically, meaning a single female can give rise to many offspring without mating, leading to explosive population growth. It was first reported in the US in 2017 when large numbers of the tick were found infesting sheep in Hunterdon County, New Jersey [88]. Enhanced surveillance during 2017–2018 subsequently detected the species in an additional eight states, and identified *H. longicornis* in archived samples from 2010 and 2013, suggesting that invasion had occurred years earlier [89]. By August 2020, established populations of *H. longicornis* had been confirmed in 14 states [90], with Ohio and Rhode Island the most recent states added to the list of

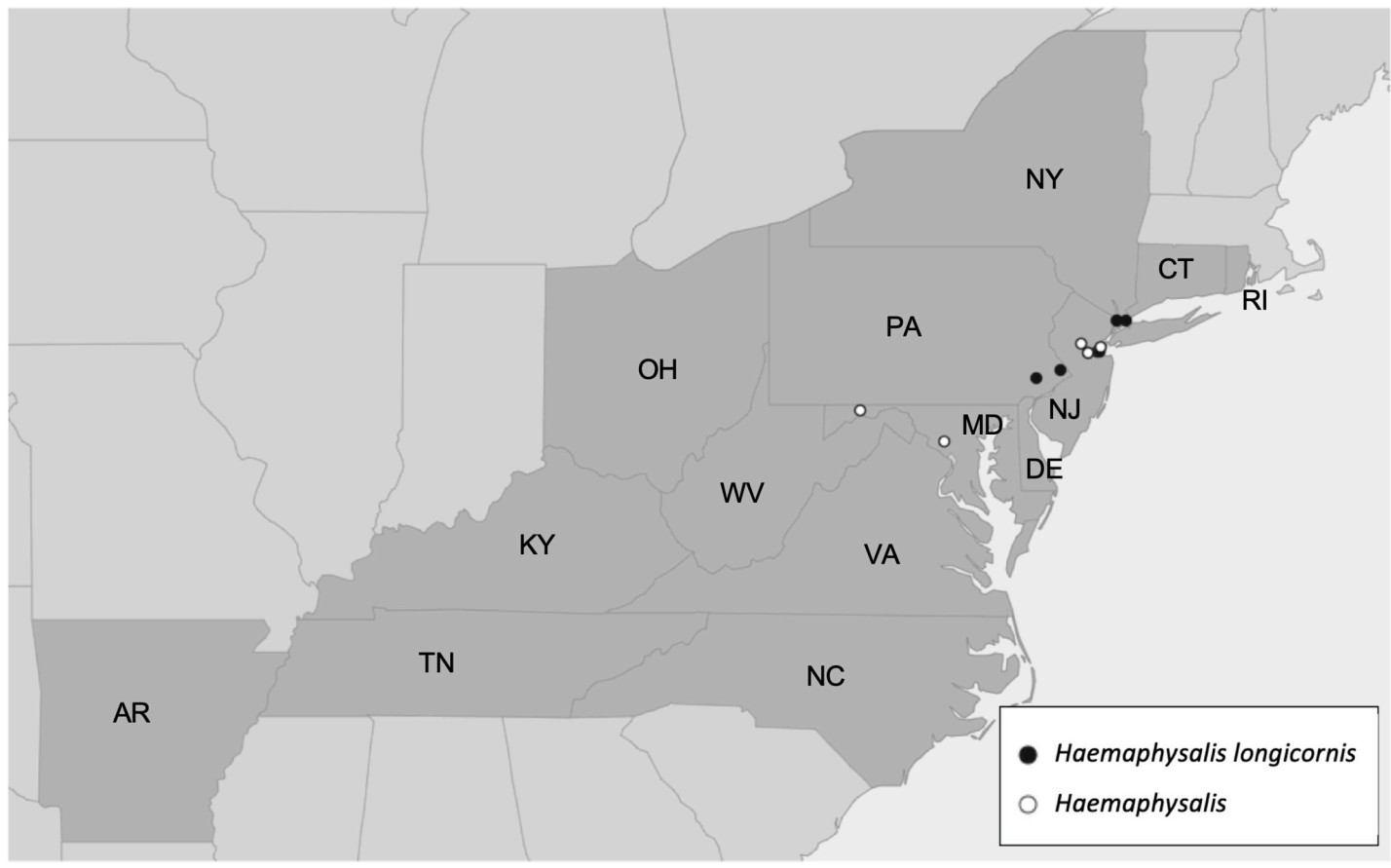

**Fig 5. Distribution of *Haemaphysalis longicornis* in the USA.** Records of *H. longicornis* (confirmed–black circles; possible–white circles) from iNaturalist. States with confirmed populations of *H. longicornis* as of 10 August 2020 are shown in dark grey with states labelled (data source: USDA, 2020). AR, Arkansas; CT, Connecticut; DE, Delaware; KY, Kentucky; MD, Maryland; NC, North Carolina; NJ, New Jersey; NY, New York; OH, Ohio; PA, Pennsylvania; RI, Rhode Island; TN, Tennessee; VA, Virginia; WV, West Virginia.

those where the tick has been found (Fig 5). Molecular barcoding of ticks from US populations suggests that there have been at least three separate introductions of *H. longicornis*, most likely from East Asia [91]. The tick has been found feeding on a variety of wild and domestic animals as well as humans [90, 92], and is a vector of important livestock pathogens, including *Theileria orientalis* which causes disease in cattle. *Theileria orientalis* genotype Ikeda was recently associated with cattle mortalities in Virginia [93] and was subsequently detected in questing *H. longicornis* collected from the same area [94]. The significance of *H. longicornis* to public health is less well understood; in its native range it is known to transmit human pathogens such as severe fever with thrombocytopenia syndrome virus and *Rickettsia japonica* (Japanese spotted fever), but it is not yet clear whether it can transmit human pathogens already present in North America. Experimental studies have so far found that it is a competent vector of *Rickettsia rickettsii* [95], but not likely to be involved in the transmission of *B. burgdorferi* [96].

The iNaturalist database was searched for *Haemaphysalis* in the USA on 31 August 2020 with a resulting 16 records found. Two records from California were excluded due to being outside the known range of *H. longicornis* in the US, and an examination of the associated images could not determine species. The remaining 14 records were from the states of New York (seven), New Jersey (three), Maryland (two) and Pennsylvania (two), all from counties with confirmed established populations of *H. longicornis*, except for one record from

Montgomery county, Maryland, where the species has not been reported to date (Fig 5). However, *Haemaphysalis* observations were only present in four of the fourteen (28.6%) states with known established *H. longicornis*. The lack of *H. longicornis* sightings throughout much of its known range may be explained by it being associated primarily with livestock and wildlife rather than humans [90], and therefore not often recorded by iNaturalist users. Five of the images associated with observations were of sufficient quality to be able to confirm ticks as *H. longicornis*, by visualization of the prominent spur on the 3$^{rd}$ palpal segment of adults, a feature that differentiates *H. longicornis* from endemic North American *Haemaphysalis* species [27]. One observation had a note from the recorder that the identification had been confirmed by the USDA National Veterinary Sciences Laboratory. The other eight observations could not be identified to species level because the photos were not taken in a manner allowing identifying features to be clearly seen.

All observations in the eastern US were recorded during May to August, with most from July (6/14; 43%) and August (5/14; 36%), consistent with peak questing activity of *H. longicornis* adults in the US [97–99]. Observations were all made in recent years with one from 2018, four from 2019, and nine from 2020, suggestive of either increasing populations of the tick and/or increasing awareness and recording.

**Case study 5: Monitoring for invasive mosquito species in Europe.** In recent decades, Europe has seen the incursion, establishment and spread of several invasive *Aedes* mosquitoes, resulting in outbreaks of mosquito-borne diseases previously only reported as imported infections [29]. Considered one of the world's top 100 invasive species by the Invasive Species Specialist Group [100], *Aedes albopictus* (Asian tiger mosquito) is the primary invasive mosquito of medical and veterinary concern. Originating in southeast Asia, this mosquito has undergone a global expansion over the last few decades, spreading to Africa, Australasia, Europe, and North and South America [29]. Its international spread has been facilitated by human activities, particularly the trade of used tyres and wet-footed plants such as "lucky bamboo" which can harbor the mosquito's cold- and drought-resistant eggs. Key to *Ae. albopictus'* invasion success is its ecological plasticity, ability to out-compete native species, and utilization of artificial water containers for breeding sites, which are readily available in urban areas [29]. It is a known vector of dengue, chikungunya, and Zika viruses, as well as *Dirofilaria* worms, and has also been shown to be a competent vector of multiple other arboviruses in experimental studies [29]. In Europe, this mosquito was first detected in 1979 in Albania, then in Italy in 1990, from where it was able to colonize most of Italy and spread eastwards and westwards along the Mediterranean coast assisted by road transport. It is currently established along the entire coast of Southern Europe and is gradually expanding northwards [101] (Fig 6A), with repeated introductions detected in Northern European countries such as the Netherlands [102], Germany [103] and the UK [104]. As a result of established populations of *Ae. albopictus*, locally-transmitted cases of chikungunya, dengue and Zika viruses have been reported in France [105, 106], dengue in Croatia [105], and two large chikungunya outbreaks have occurred in Italy [107].

Other invasive *Aedes* species established and spreading in Europe are *Ae. japonicus* and *Ae. koreicus* [29]. Whilst neither species is considered an important disease vector in their native ranges, *Ae. japonicus* is a competent vector of several viruses in the laboratory and there are concerns that it could become involved in West Nile virus transmission in Europe [29]. *Aedes koreicus* has been implicated in the transmission of *Dirofilaria immitis* and Japanese encephalitis virus. Both mosquito species readily bite humans and can therefore cause considerable nuisance biting. There appear to have been multiple introductions of *Ae. japonicus* and this species is now established widely in Central European countries [108] (Fig 6B), whereas *Ae. koreicus* occurs in a few localized populations in Central Europe as well as in Russia (Fig 6C).

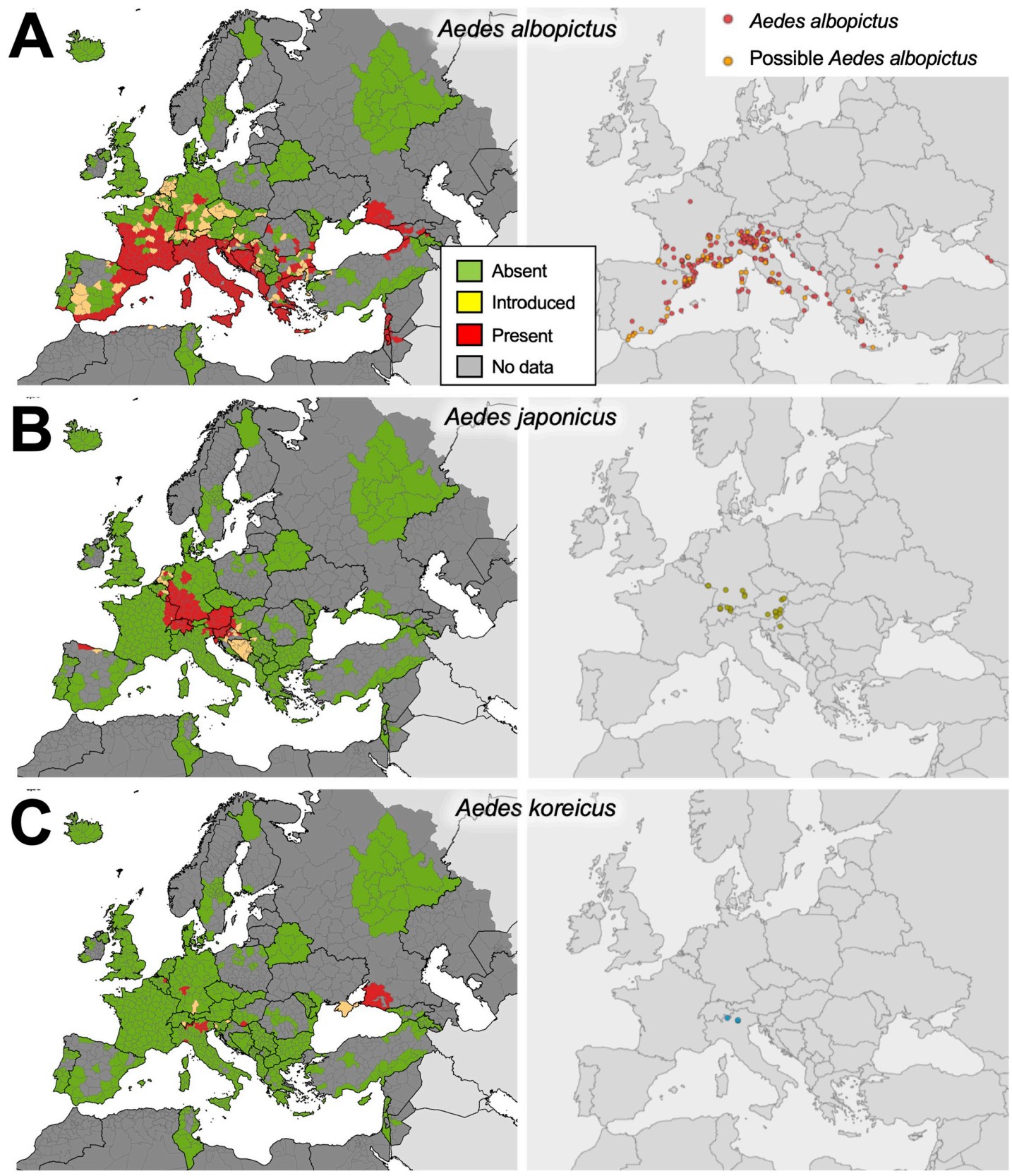

**Fig 6. Distribution of invasive *Aedes* mosquito records in Europe and neighboring territories.** Known distribution of (A) *Aedes albopictus*, (B) *Aedes japonicus*, and (C) *Aedes koreicus* in Europe as of May 2020 (left panels; reprinted from European Centre for Disease Control and Prevention VectorNet project www.ecdc.europa.eu/ en/disease-vectors/surveillance-and-disease-data/mosquito-maps, original copyright ECDC [2020]), and distribution of observations of *Aedes albopictus*, *Aedes japonicus* and *Aedes koreicus* in Europe from iNaturalist (right panels).

*Aedes aegypti* is also a species of concern for Europe; this mosquito is an important disease vector globally, and responsible for the majority of dengue, chikungunya, Zika and yellow fever cases. Although once present in southern Europe, *Ae. aegypti* was eliminated from much of the continent, but has recently recolonized the Portuguese autonomous region of Madeira and the Black Sea coast of Russia, Georgia and Turkey [29, 101] and has also been intercepted in the Netherlands following introduction via international air traffic [109]. Following its establishment on Madeira in 2004–2005, *Ae. aegypti* was associated with an outbreak of dengue with over 2000 cases [105]. As much of southern Europe is climatically suitable for *Ae. aegypti* survival, there is potential for the mosquito to become re-established and, along with *Ae. albopictus*, facilitate further arboviral transmission in the region [29].

The iNaturalist database was searched on 31st July 2020 for observations of *Aedes aegypti*, *Aedes albopictus*, *Aedes japonicus*, and *Aedes koreicus* in Europe. Only two records were found in the *Aedes aegypti* search: one was a misidentified native *Aedes* species in Germany, and the other specimen was too damaged to identify conclusively, although it was black and white scaled and reported in the species' known range on the Black Sea coast of Russia, where *Ae. albopictus* is also present.

The search for *Aedes albopictus* came up with 315 records. Of these, two were not mosquitoes, and nine were identified as other *Aedes* species. Six records were photos of mosquito larvae but did not include images of all the features required to identify to species level. Of the remaining 298 observations, 212 (71.1%) could be confidently identified as *Ae. albopictus* based on visualization of bright white and black scales, banded tarsi and a wide medial stripe on the scutum, whereas 86 (28.9%) were classed as 'possible *Aedes albopictus*' because they had black and white scales and banded tarsi but the scutum could not be clearly seen to confirm the species. In countries where the morphologically similar *Aedes cretinus* is present (i.e. Greece, Cyprus, Turkey and Georgia), care was taken to look at the posterior dorsocentral stripes on the scutum, a feature that can differentiate these species [28]. Observations of *Ae. albopictus* (confirmed and possible) were made in 14 countries (Albania, Bulgaria, Croatia, France, Gibraltar, Greece, Italy, Montenegro, Romania, Russia, Slovenia, Spain, Switzerland and Turkey), with most records reported in Italy (136/298; 45.6%), France (86/298; 28.9%), and Spain (46/298; 15.4%), and appearing to cluster around major urban centers such as Rome, Lyon, Barcelona and Athens (Fig 6A). Consistent with this species' known distribution, records were widespread along the Mediterranean coast, and in the northern provinces of Italy, but there were no records outside its known range. The species was recorded as far north as Paris (Fig 6A). *Aedes albopictus* were recorded from February to November, with most observations May to October and peak numbers during August and September (whether comparing confirmed *Ae. albopictus* only, or confirmed plus possible *Ae. albopictus*), in agreement with known seasonality of the species in Europe [29].

Twenty-three observations of *Aedes japonicus* were found, and all could be confirmed by the patterns on the scutum and tarsi (i.e. presence of longitudinal lines of yellowish scales on the scutum, and banded tarsi with basal bands on hind tarsomeres 1–3). All records were within the known ranges of the mosquito in Europe (Fig 6B) with eleven observations in Austria, six in Germany, five in Switzerland and one in Croatia. *Aedes japonicus* was recorded April to October with most records in July.

There were three records of *Aedes koreicus*, all of which could be confirmed by visualization of the patterns on the scutum and banding pattern on the tarsi (scutum similar to *Ae. japonicus* but with complete basal bands on hind tarsomeres 1–4), and all were in northern Italy within this species' known established range in Europe (Fig 6C).

## Discussion

The aim of this study was to explore the availability of observations of important vector taxa on the iNaturalist platform and determine the utility of these data to complement existing vector surveillance activities. Using examples of ticks and mosquitoes in regions of North America and Europe (according to the author's areas of expertise), this study shows that data from iNaturalist can be a valuable source of additional information on vector distribution and seasonality and can provide extra evidence of the introduction and expansion of key tick species that could be used to support risk assessments in affected areas. It could be especially useful at a time when many vector surveillance programs may have been interrupted because of COVID-19 lockdowns, travel restrictions and redistribution of public health agency resources to the COVID-19 response, combined with a potentially higher number of observations made by citizens who have spent more time than usual recreating outdoors because of interruptions to regular activities. However, limitations of this work are that case studies only focused on ticks and mosquitoes, the most represented vector taxa in iNaturalist, in areas with the most iNaturalist users (Europe and North America), and therefore further analysis would be needed to determine whether iNaturalist data would support surveillance for other vector groups and on other continents.

Tick data collected at the national or state level (case studies 1 and 2) were sufficient to determine seasonal occurrence and distribution patterns of tick species of importance, despite being based on relatively few observations. Whilst the coverage of iNaturalist records could not compete with existing data, which have been collected over decades of active and passive surveillance, they were able to corroborate and complement known trends in tick distribution. Importantly, in both case studies tick species with expanding distributions (i.e. *H. punctata* and *Dermacentor* in the UK, and *A. americanum* in Minnesota) were detected, as well as sightings of less commonly encountered species such as *I. frontalis* and *C. kelleyi*.

Using data from iNaturalist to monitor expanding ranges of tick species in Europe (*Hyalomma*) and the US (*H. longicornis*), and invasive *Aedes* mosquitoes in Europe, also showed potential for tracking these species in their known range as well as identifying possible areas of expansion, and this approach could be applied to other vectors and regions such as *I. scapularis* expansion in Canada, *A. americanum* expansion in North America, the expansion of *I. ricinus* at the northern edges of its range in Europe, and recording observations of invasive *Aedes* mosquitoes in other temperate areas of the globe. However, these case studies also highlight the shortcomings of using iNaturalist in the absence of traditional surveillance practices. For example, whilst *H. longicornis* was observed in a county in which its establishment has so far not been reported, there were no iNaturalist records of this tick species in 10 of the states where known populations have been detected by traditional surveillance approaches.

There are many limitations that are associated with using data from crowdsourced records, which are common to many forms of citizen science used in vector surveillance [110]. Firstly, data are based on opportunistic sightings of wildlife, and are spatially and temporally biased to where people live [2] (e.g. see case study 2) and when and where they spend time outdoors, as well as potentially being biased to the types of wildlife users are interested in. Mueller *et al.* (2019) found that rather than mapping true urban canid distribution, iNaturalist recorded human-urban canid interactions, and similarly for vector surveillance this collection method

may be more representative of human exposure to vectors rather than to true vector distribution. The case studies above suggest that iNaturalist is more likely to be able to capture vector species commonly encountered by humans and their companion animals than those associated with livestock or wildlife (e.g. *H. longicornis*). As with other passive surveillance approaches, this method of data collection only captures presence and not absence data. Another potential problem with the location data is mapping issues, leading to incorrect reporting of a species' geographical occurrence. This can be a particular problem for tick records, as in some cases the location the tick is found and photographed is different to where the tick was acquired. For example, two observations in the UK dataset included notes on where the tick bite had occurred, yet both records were georeferenced to the presumably hometown location where the ticks were found and recorded, both many miles from where the tick was thought to have been acquired. This is a well-known issue encountered during passive surveillance for ectoparasites that may travel on their host before being detected and removed [110]. Importantly, no follow up investigations were carried out during this study, so it could not be confirmed whether the tick species detected outside their normal ranges (i.e. *Dermacentor* in the UK, *A. americanum* in Minnesota, and *Hyalomma* in Czechia) were locally acquired/ observed, and thus genuine records of expansion/invasion, or associated with travel to endemic regions. Furthermore, iNaturalist observations of vectors also tend to lack associated data that are usually collected by passive vector surveillance systems, such as host association (although this is occasionally included or can be observed in the image), or any health effects in the host, and of course the specimen itself is not available for further analyses such as pathogen screening or confirmation of identification by molecular or morphological means.

Observations are primarily photographic observations in the field and usually not amenable to identification through morphological keys, therefore identification of many species requires considerable expertise, experience and knowledge of the local/regional vector situation. As noted recently in a review of approaches of citizen science for collecting tick data, the identification of vectors from digital images is highly sensitive to both image quality and the expertise of the individual identifying the images [110]. Two recent studies examining the accuracy of photographic identification when compared to morphological and/or molecular identification of matched tick samples found that when images were examined by experienced entomologists there was a high accuracy of identification [111, 112], supporting the use of image-based identification for vector monitoring. One of these studies, performed in the US, reported an overall accuracy of photo-based identification of 96.7%, and accuracy was 98–99% for the three most common tick species of medical importance, *I. scapularis*, *D. variabilis* and *A. americanum* [112]. Successful identification was not significantly affected by tick engorgement, season, or location of acquisition, but was affected by tick species and life-stage, with more common species more easily identified and nymphs and larvae more difficult to correctly identify in images than adults [112]. The second study, on ticks submitted by veterinary practices in Canada, had an image-based identification accuracy of 97.2%, although this was after 26% of submitted images had been excluded due to not being of suitable quality to allow identification [111]. Both of these studies provided participants with guidelines for photographing ticks to ensure that high quality images with the key anatomical features visible were submitted [111, 112]. As many iNaturalist recorders may be unaware of the key features required for vector identification and because images are primarily captured with smartphones, the resulting identification accuracy rate can be expected to be slightly lower than the studies cited above, primarily because of the need to exclude low quality images. This image-based method of observation may lend itself to the identification of certain vector groups, such as ticks and triatomines, which are relatively large and slow-moving compared to mosquitoes and midges for example, which are comparatively more difficult to photograph whilst capturing the key features

required for identification. It is also better suited to identification of distinctive species with easily recognizable features such as *A. americanum* (the "Lonestar" marking on the scutum of females) and *Ae. albopictus* (distinctive white line on middle of scutum and banded tarsi). One limitation of the searches performed in this study is the specificity of the search term, as using a species or genus (or even a higher taxon) will likely result in the exclusion of many records that have not yet been identified or have been misidentified as other species, families, or even orders.

Taking these limitations into consideration, it is suggested that iNaturalist be used as a complementary approach to existing local vector surveillance, with observations regularly monitored and any interesting records followed up and confirmed by encouraging iNaturalist users to send in specimens to local vector identification services. Adding iNaturalist searches to an existing combination of passive and active surveillance approaches will ensure that the maximum amount of data is utilized in monitoring and risk assessment for arthropod vectors of concern. One benefit of iNaturalist over existing passive vector surveillance schemes and smartphone apps (apart from its extremely low cost) is that users do not need to have any awareness of vectors before submitting their observations, making it possible for citizens unaware of vectors to contribute and also offers an opportunity to improve their knowledge on potential vector borne disease risks. It is also worth noting that other citizen-sourced images on social media platforms, such as Twitter, Facebook, Instagram and Flickr, could provide additional records for vector surveillance, and some of these sources have already been used by researchers to obtain biodiversity data [18, 113, 114].

Entomologists with taxonomic expertise are encouraged to engage with the iNaturalist platform and use their knowledge to identify observations and improve the vector data available to the community. The database is also a powerful teaching and training tool, and photos from the many observations can be used to train both scientists and citizens in the recognition of key vector genera and species. With the increasing use of smartphones and the growing number of users and observations on iNaturalist, the data look set to keep increasing into the future and may prove a valuable resource to assist vector surveillance. Additionally, the rapid development of smartphone technology, particularly in the quality of smartphone cameras, should lead to facilitation of species identification in future observations.

## Supporting information

**S1 Fig. Example images of vectors.** (a) case study 1, *Ixodes ricinus* recorded in the UK; (b) case study 2, *Amblyomma americanum* recorded in Minnesota, USA; (c) case study 4, record of *Haemaphysalis longicornis* from New York state, USA; (d) case study 3, *Hyalomma* spp. recorded in Spain; (e) case study 5, *Aedes albopictus* observed in Italy; (f) case study 5, *Aedes japonicus* observed in Austria. These images were taken by iNaturalist users and are made available under Creative Commons license (CC BY 4.0).
(TIF)

**S1 Table.**
(XLSX)

## Acknowledgments

I would like to thank the iNaturalist community for their observations of arthropod vector species, particularly those users whose records were included in this analysis. I am grateful to Emma Gillingham (Public Health England, UK), Jonathan Oliver (School of Public Health, University of Minnesota) and Ulrike Munderloh (Department of Entomology, University of Minnesota) for comments on the manuscript.

## Author Contributions

**Conceptualization:** Benjamin Cull.

**Data curation:** Benjamin Cull.

**Formal analysis:** Benjamin Cull.

**Investigation:** Benjamin Cull.

**Methodology:** Benjamin Cull.

**Project administration:** Benjamin Cull.

**Validation:** Benjamin Cull.

**Visualization:** Benjamin Cull.

**Writing – original draft:** Benjamin Cull.

**Writing – review & editing:** Benjamin Cull.

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
