## [Decision Letter · Decision Letter 0]

12 Jan 2021

PONE-D-20-33287

Potential for online crowdsourced biological recording data to complement surveillance for arthropod vectors

PLOS ONE

Dear Dr. Cull,

Thank you very much for submitting your manuscript, "Potential for online crowdsourced biological recording data to complement surveillance for arthropod vectors" (PONE-D-20-33287), for consideration at PLOS ONE. As with all papers reviewed by the journal, your manuscript was reviewed by members of the editorial board and by several independent reviewers. In light of the reviews (below this email), we would like to invite the resubmission of a significantly-revised version that takes into account the reviewers' comments.

We look forward to receiving your revised manuscript.

Kind regards,

Abdallah M. Samy, PhD

Academic Editor

PLOS ONE

**Journal Requirements:**

3. We note that Figure(s) S1, 2, 3, 4, 5 and 6 in your submission contain map/satellite images which may be copyrighted. All PLOS content is published under the Creative Commons Attribution License (CC BY 4.0), which means that the manuscript, images, and Supporting Information files will be freely available online, and any third party is permitted to access, download, copy, distribute, and use these materials in any way, even commercially, with proper attribution. For these reasons, we cannot publish previously copyrighted maps or satellite images created using proprietary data, such as Google software (Google Maps, Street View, and Earth). For more information, see our copyright guidelines: http://journals.plos.org/plosone/s/licenses-and-copyright.

a.    You may seek permission from the original copyright holder of Figure(s) S1, 2, 3, 4, 5 and 6 to publish the content specifically under the CC BY 4.0 license. 

**Additional Editor Comments:**

Please address all reviewer comments below before submitting a revised version of this manuscript. I have to unassign any additional reviewer on this manuscript to avoid any further delay. 

**Reviewers' comments:**

Reviewer's Responses to Questions

**Comments to the Author**

1. Is the manuscript technically sound, and do the data support the conclusions?

Reviewer #1: No

2. Has the statistical analysis been performed appropriately and rigorously? 

Reviewer #1: N/A

3. Have the authors made all data underlying the findings in their manuscript fully available?

Reviewer #1: Yes

4. Is the manuscript presented in an intelligible fashion and written in standard English?

Reviewer #1: Yes

5. Review Comments to the Author

Reviewer #1: According to the author of this manuscript, large datasets are collected voluntarily by citizen scientists over wide geographical areas that are increasingly utilized by researchers for multiple applications, including arthropod vector surveillance. Among the different sources of these datasets, the iNaturalist is described as an on-line platform that accumulate crowdsourced biological observations from around the world that could also be useful for monitoring vectors. Therefore, the aim of this study was to assess the availability of observations of important vector taxa on the iNaturalist platform and to examine the utility of these data to complement existing vector surveillance activities. As described in the Abstract, the results based on examining case studies using observations from the iNaturalist led to the claim that crowdsourced data from the online database iNaturalist can supplement existing vector surveillance information. However, the claim based on the contents of the abstract is not supported by an adequate description of plans and methods that were used to assess the iNaturalist as a source of information and utility of the data pertaining to vector surveillance. Therefore, the abstract could be improved by including a brief description of the plan and methods used to assess the availability of vector observations and to examine the utility of the observations to complement vector surveillance. Also, the abstract could be improved by including the results for the aim to assess the availability of observations of important vector taxa in the on-line iNaturalist platform.

The Introduction present an excellent description of the use of citizen scientist to employ a variety of methods to collect scientific data, including vector surveillance data on arthropod species distribution, seasonality and human/animal exposure. Among these methods, the scope of the iNaturalist is described as a platform that has been expanded to include a rapidly growing repository of biological records, but has not yet been fully explored as a source of information for studying vector species of importance to public and veterinary health. Therefore, the rationale for this study was to further explore the use of the iNaturalist as a source of information to complement existing vector surveillance activities. However in Section B of Vector surveillance at a national or regional level, it is stated that “The potential for data from iNaturalist to be used for vector surveillance at a country or state level was assessed” rather than examined as stated previously stated in the abstract and introductions. While the contents of the introduction section is relevant to the aims of this study, the rationale could be made clearer for why the iNaturalist’s platform was selected for assessment as a source of vector surveillance information. Were the cases studies the only studies in the iNaturalist of relevance to vector surveillance? What advantages, if any is the iNaturalist platform for collecting information over other sources of citizen science observations for supplementing vector surveillance activities. That is, what is known about other sources of citizen scientist vector surveillance information? How widely known is the iNaturalist as a platform for up-loading information of relevance to vector surveillance? What are the advantages, if any of using citizen science observations over published information. Furthermore, would a survey of personnel involved in vector surveillance programs be a more objective assessment of the availability and use of the iNaturalist platform as a source of information to compliment vector surveillance?

As alluded to by the reviewer’s comments on the abstract, the methods are not adequately described to perform an objective and targeted assessment of the availability of data that extends beyond only performing an estimate of the number of observations and to examine the use of the data based on selected case studies. Thus, the methods are incomplete and lacking in sufficient detail to understand how the results for each aim was achieved. For example, what does important vector taxa encompass, does this include all known vector families, genera and species? Of the records that were found on the iNaturalist, how many pertained to a vector species? For example, the number of records were presented but what was the criteria used to determine how many records were considered relevant to vector surveillance. Evidence based on the search for a few selected species of interest to perform case studies is not sufficient to make an objective and overall assessment of the use of iNaturalist platform of relevance to vector surveillance information. What methods were used to assess why, when and how the observations were made by citizen scientists of relevance to vector surveillance other than that images were taken of vectors and uploaded in the iNaturalist platform. What were the methods used and how were they assess/examined to determine if the information was appropriate for use to provide temporal and spatial information needed to generate to supplementing vector surveillance program, especially in view of the duplication of existing information. Overall, a plan with criteria and defined methods presented in the Method section to address the aims would have provided a more comprehensive objective assessment and examination of the iNaturalist as a possible source of information that could contribute effectively as a supplement to vector surveillance.

• The results and discussion based on records of the availability of data provided information that supports the iNaturalist platform as a source vector surveillance information. However, the actual practical value of the information regarding the support of vector surveillance is not clear and is an important gap in the assessment of the iNaturalist as a source of information. For example, the case studies demonstrated that the data collected provided supporting evidence of the possible utility of the iNaturalist data, but it is not clear if any of the observations were needed to supplement existing vector surveillance information. These weaknesses are further evidence of the lack of a comprehensive design in the approach and methods for making an overall assessment of the iNaturalist information system for supporting vector surveillance activities. As mentioned earlier in this review, the author did not examine the iNaturalist platform for duplication of existing information, which is another gap in the assessment of the need and/or use of the information. Furthermore, it is stated that iNaturalist information could be useful in areas where vector surveillance programs may be disrupted by COVID-19 lockdowns or where vector surveillance programs do not exist. With the very limited information that is uploaded by senior scientist to the iNaturalist platform, including only the possibility to identify and understand the seasonal distribution of vectors, such information alone will not suffice to support a vector surveillance program that does not exist. The author present an excellent analysis of the limitations of the iNaturalist as a source of vector surveillance data that overlaps in-part with that of the reviewer’s observations. Identification of these limitations is a very important contribution, and if used to resolved these gaps, this may lead to substantial improvements in the iNaturalist platform as a source of information for supplementing vector surveillance activities. Overall, based on the author’s assessment and examination of the iNaturalist as a source of vector surveillance data, the limited scope of the approach and in-complete methods precluded the performance a comprehensive coverage needed to assess the INaturalist as a source of information for complimenting vector surveillance activities.

6. PLOS authors have the option to publish the peer review history of their article (what does this mean?). If published, this will include your full peer review and any attached files.

Reviewer #1: No

---

## [Author Response · Author response to Decision Letter 0]

1 Feb 2021

Journal Requirements:

> The manuscript has been formatted to match the journal style and files renamed. I apologise for overlooking these requirements during submission.

> This phrase “data not shown” has been removed from the manuscript without affecting the results.

3. We note that Figure(s) S1, 2, 3, 4, 5 and 6 in your submission contain map/satellite images which may be copyrighted. All PLOS content is published under the Creative Commons Attribution License (CC BY 4.0), which means that the manuscript, images, and Supporting Information files will be freely available online, and any third party is permitted to access, download, copy, distribute, and use these materials in any way, even commercially, with proper attribution. For these reasons, we cannot publish previously copyrighted maps or satellite images created using proprietary data, such as Google software (Google Maps, Street View, and Earth). For more information, see our copyright guidelines: http://journals.plos.org/plosone/s/licenses-and-copyright.

a. You may seek permission from the original copyright holder of Figure(s) S1, 2, 3, 4, 5 and 6 to publish the content specifically under the CC BY 4.0 license. 

> The map data has been removed from S1 Fig images without affecting the data, since this figure is only intended to provide examples of photographs of vectors.

The other map figures have been remade using open source map layers from Natural Earth. The images in Figs 4 & 6 from ECDC are available for reuse under a license consistent with CC BY 4.0 - please find attached details of the license provided by ECDC and correspondence with PLOS ONE publications staff confirming that these are compliant with CC BY 4.0 sharing. 

 

5. Review Comments to the Author

Reviewer #1: According to the author of this manuscript, large datasets are collected voluntarily by citizen scientists over wide geographical areas that are increasingly utilized by researchers for multiple applications, including arthropod vector surveillance. Among the different sources of these datasets, the iNaturalist is described as an on-line platform that accumulate crowdsourced biological observations from around the world that could also be useful for monitoring vectors. Therefore, the aim of this study was to assess the availability of observations of important vector taxa on the iNaturalist platform and to examine the utility of these data to complement existing vector surveillance activities. As described in the Abstract, the results based on examining case studies using observations from the iNaturalist led to the claim that crowdsourced data from the online database iNaturalist can supplement existing vector surveillance information. However, the claim based on the contents of the abstract is not supported by an adequate description of plans and methods that were used to assess the iNaturalist as a source of information and utility of the data pertaining to vector surveillance. Therefore, the abstract could be improved by including a brief description of the plan and methods used to assess the availability of vector observations and to examine the utility of the observations to complement vector surveillance. Also, the abstract could be improved by including the results for the aim to assess the availability of observations of important vector taxa in the on-line iNaturalist platform.

> The abstract has been modified to include an overview of the methods and a summary of the results from the exploration of available records of important vector taxa.

The Introduction present an excellent description of the use of citizen scientist to employ a variety of methods to collect scientific data, including vector surveillance data on arthropod species distribution, seasonality and human/animal exposure. Among these methods, the scope of the iNaturalist is described as a platform that has been expanded to include a rapidly growing repository of biological records, but has not yet been fully explored as a source of information for studying vector species of importance to public and veterinary health. Therefore, the rationale for this study was to further explore the use of the iNaturalist as a source of information to complement existing vector surveillance activities. However in Section B of Vector surveillance at a national or regional level, it is stated that “The potential for data from iNaturalist to be used for vector surveillance at a country or state level was assessed” rather than examined as stated previously stated in the abstract and introductions. 

> In section B of Results, the text has been corrected to make it clear that the data are being assessed as a source of additional records for vector surveillance, rather than to use iNaturalist data alone for surveillance. Thanks for pointing this out.

While the contents of the introduction section is relevant to the aims of this study, the rationale could be made clearer for why the iNaturalist’s platform was selected for assessment as a source of vector surveillance information. Were the cases studies the only studies in the iNaturalist of relevance to vector surveillance? What advantages, if any is the iNaturalist platform for collecting information over other sources of citizen science observations for supplementing vector surveillance activities. That is, what is known about other sources of citizen scientist vector surveillance information? How widely known is the iNaturalist as a platform for up-loading information of relevance to vector surveillance? What are the advantages, if any of using citizen science observations over published information. Furthermore, would a survey of personnel involved in vector surveillance programs be a more objective assessment of the availability and use of the iNaturalist platform as a source of information to compliment vector surveillance?

> Some additional sentences have been added to the introduction to add more background information on citizen science surveillance and improve clarity to the rationale behind investigating the use and potential advantages of iNaturalist in vector surveillance. 

Part of the reason for starting this study was to raise awareness of iNaturalist as a possible additional source of data for vector surveillance, as the case studies reveal that it could add records of invasive species which might be important evidence for expansion, which is useful data for those monitoring invasive species. 

I expect that the awareness of iNaturalist among the vector surveillance community is low, although I know of a couple of individuals at state health departments that are using it to collect additional records of ticks in their regions. Therefore, I have added a sentence to highlight this aim. 

A survey of vector surveillance personnel is a great idea, but I would want to have some evidence showing the potential use of iNaturalist data before contacting them, hence these case studies were conducted. These could be used as examples of how it could work, and personnel could then decide whether it would aid their current vector surveillance activities.

There are many possible relevant case studies that could be examined using iNaturalist data, but these are the only five that I performed. They were chosen based on my expertise in ticks and key invasive species of ticks and mosquitoes, and because these are the most important vector groups in terms of disease transmission worldwide. All five of the studies I performed suggested that iNaturalist would be useful for supplementing surveillance through the addition of extra records and four of the five detected new records of expanding species.

As alluded to by the reviewer’s comments on the abstract, the methods are not adequately described to perform an objective and targeted assessment of the availability of data that extends beyond only performing an estimate of the number of observations and to examine the use of the data based on selected case studies. Thus, the methods are incomplete and lacking in sufficient detail to understand how the results for each aim was achieved. For example, what does important vector taxa encompass, does this include all known vector families, genera and species? Of the records that were found on the iNaturalist, how many pertained to a vector species? For example, the number of records were presented but what was the criteria used to determine how many records were considered relevant to vector surveillance. Evidence based on the search for a few selected species of interest to perform case studies is not sufficient to make an objective and overall assessment of the use of iNaturalist platform of relevance to vector surveillance information. What methods were used to assess why, when and how the observations were made by citizen scientists of relevance to vector surveillance other than that images were taken of vectors and uploaded in the iNaturalist platform. What were the methods used and how were they assess/examined to determine if the information was appropriate for use to provide temporal and spatial information needed to generate to supplementing vector surveillance program, especially in view of the duplication of existing information. Overall, a plan with criteria and defined methods presented in the Method section to address the aims would have provided a more comprehensive objective assessment and examination of the iNaturalist as a possible source of information that could contribute effectively as a supplement to vector surveillance.

> The methods section has been expanded to include a detailed description of the methods and criteria used to assess the iNaturalist data in comparison to existing information from vector surveillance. 

Because there are so many case studies that could be performed and the species of interest will vary by region and profession etc. this can only be an incomplete assessment and is meant to highlight the potential for iNaturalist as a tool to complement existing vector surveillance data. I have added some lines to the discussion to highlight that other assessments would need to be done in different regions and with different vector taxa in order to see how iNaturalist performs in other case studies.

• The results and discussion based on records of the availability of data provided information that supports the iNaturalist platform as a source vector surveillance information. However, the actual practical value of the information regarding the support of vector surveillance is not clear and is an important gap in the assessment of the iNaturalist as a source of information. For example, the case studies demonstrated that the data collected provided supporting evidence of the possible utility of the iNaturalist data, but it is not clear if any of the observations were needed to supplement existing vector surveillance information. These weaknesses are further evidence of the lack of a comprehensive design in the approach and methods for making an overall assessment of the iNaturalist information system for supporting vector surveillance activities. As mentioned earlier in this review, the author did not examine the iNaturalist platform for duplication of existing information, which is another gap in the assessment of the need and/or use of the information. Furthermore, it is stated that iNaturalist information could be useful in areas where vector surveillance programs may be disrupted by COVID-19 lockdowns or where vector surveillance programs do not exist. With the very limited information that is uploaded by senior scientist to the iNaturalist platform, including only the possibility to identify and understand the seasonal distribution of vectors, such information alone will not suffice to support a vector surveillance program that does not exist. The author present an excellent analysis of the limitations of the iNaturalist as a source of vector surveillance data that overlaps in-part with that of the reviewer’s observations. Identification of these limitations is a very important contribution, and if used to resolved these gaps, this may lead to substantial improvements in the iNaturalist platform as a source of information for supplementing vector surveillance activities. Overall, based on the author’s assessment and examination of the iNaturalist as a source of vector surveillance data, the limited scope of the approach and in-complete methods precluded the performance a comprehensive coverage needed to assess the INaturalist as a source of information for complimenting vector surveillance activities.

> The iNaturalist data can provide supporting information and additional records to existing surveillance data which is useful evidence that can be used to support regional risk assessments performed by public health agencies for example. A line on this has been added to the discussion.

The data from iNaturalist were compared with existing data from literature and government agencies, and any duplication (matching existing data) is described in the text and can be seen in the figures. Duplication of existing information means that iNaturalist data is supportive of the known distribution, seasonality etc. so this is a good sign that the crowdsourced data are reliable, and that additional records outside known ranges of vectors can be trusted. This has been added to the methods.

The line on iNaturalist use in areas without existing vector surveillance has been deleted, although data in this study suggest that iNaturalist data provide a good indication of species present and their seasonality comparable to known data, so in a low-resource setting iNaturalist could be a good place to start if you have no existing data.

---

## [Editor Report · Decision Letter 1]

6 Apr 2021

Potential for online crowdsourced biological recording data to complement surveillance for arthropod vectors

PONE-D-20-33287R1

Dear Dr. Cull,

We’re pleased to inform you that your manuscript, "Potential for online crowdsourced biological recording data to complement surveillance for arthropod vectors" (PONE-D-20-33287R1), has been judged scientifically suitable for publication and will be formally accepted for publication once it meets all outstanding technical requirements.

Kind regards,

Abdallah M. Samy, PhD

Academic Editor

PLOS ONE

---

## [Editor Report · Acceptance letter]

21 Apr 2021

PONE-D-20-33287R1 

Potential for online crowdsourced biological recording data to complement surveillance for arthropod vectors 

Dear Dr. Cull:

I'm pleased to inform you that your manuscript has been deemed suitable for publication in PLOS ONE. Congratulations! Your manuscript is now with our production department. 

Kind regards, 

on behalf of

Dr. Abdallah M. Samy 

Academic Editor

PLOS ONE